# Early assessment of fungal and oomycete pathogens in greenhouse irrigation water using Oxford nanopore amplicon sequencing

Enoch Narh Kudjordjie[1]*, Anne Saaby Schmidt-Høier[2], Mai-Britt Brøndum[2], Mads Grønvald Johnsen[2], Mogens Nicolaisen[1], Mette Vestergård[1]

1 Department of Agroecology, Faculty of Technical Sciences, Aarhus University, Slagelse, Denmark,
2 Teknologisk Institut Gregersensvej, Taastrup, Denmark

* enochnarh@agro.au.dk

## Abstract

Water-borne plant pathogenic fungi and oomycetes are a major threat in greenhouse production systems. Early detection and quantification of these pathogens would enable us to ascertain both economic and biological thresholds required for a timely treatment, thus improving effective disease management. Here, we used Oxford nanopore MinION amplicon sequencing to analyze microbial communities in irrigation water collected from greenhouses used for growing tomato, cucumber and *Aeschynanthus* sp. Fungal and oomycete communities were characterized using primers that amplify the full internal transcribed spacer (ITS) region. To assess the sensitivity of the MinION sequencing, we spiked serially diluted mock DNA into the DNA isolated from greenhouse water samples prior to library preparation. Relative abundances of fungal and oomycete reads were distinct in the greenhouse irrigation water samples and in water samples from setups with tomato that was inoculated with *Fusarium oxysporum*. Sequence reads derived from fungal and oomycete mock communities were proportionate in the respective serial dilution samples, thus confirming the suitability of MinION amplicon sequencing for environmental monitoring. By using spike-ins as standards to test the reliability of quantification using the MinION, we found that the detection of spike-ins was highly affected by the background quantities of fungal or oomycete DNA in the sample. We observed that spike-ins having shorter length (538bp) produced reads across most of our dilutions compared to the longer spikes (>790bp). Moreover, the sequence reads were uneven with respect to dilution series and were least retrievable in the background samples having the highest DNA concentration, suggesting a narrow dynamic range of performance. We suggest continuous benchmarking of the MinION sequencing to improve quantitative metabarcoding efforts for rapid plant disease diagnostic and monitoring in the future.

## Introduction

Plant production in greenhouse and in vertical farming systems is increasing, primarily due to its potential for improving crop health and yields, as well as circumventing high crop yield

**Data Availability Statement:** Sequence data are available in the Sequence Read Archive (SRA) database at NCBI under BioProject ID

PRJNA1063628 with the accession numbers SUB14149061, SUB14149186, SUB14152390, SUB14152424, SUB14144130 and SUB14152446.

**Funding:** The project was funded by the Grønt Udviklings-og DemonstrationsProgram (GUDP) project no. 32002. The funders had no role in study design, data collection and analysis, decision to publish, or preparation of the manuscript.

**Competing interests:** The authors declare no conflict of interest.

losses in open fields due to stresses caused by climate change [1]. However, such intensive cropping systems are exceptionally conducive for the development of plant pathogens. One of the primary pathways for pathogen dispersal in greenhouse systems is the irrigation water, a major substrate of pathogen growth and reproduction, dispersal and spread of diseases [2,3]. Fungal and oomycete pathogens are the causative agents of most diseases in greenhouse systems resulting in huge economic losses. Fungal pathogens within the genera *Fusarium*, mostly *F. oxysporum*, are notably known for causing wilt, rot and blight diseases [4]. Oomycetes, particularly *Phytophthora* and *Pythium* species are zoosporic organisms that are well adapted to the aquatic environment, and are a huge threat to many crops, causing stem and root rot, fruit rot, shoot cankers, damping off diseases and foliar blight [4,5].

Water-borne fungal and oomycete pathogens have been reported to survive and easily spread through recirculating systems [4,6,7], while irrigation water quality as well as the type of irrigation, cropping density and water treatment affect inoculum loads and species diversity in irrigation systems [8]. Previous reports have revealed a strong correlation between increased disease incidence and recycled water usage [9].

Presently, there is an increasing demand for water recycling in greenhouse systems. In the European Union, regulation on minimum requirements of water reuse for agricultural irrigation is enforced from 2023 (https://ec.europa.eu/environment/water/reuse.htm), which is expected to further encourage and facilitate water reuse in greenhouse systems in the European countries. While this policy will improve water use efficiency, recycled water could contain high numbers of infective propagules of pathogens, and thus increase risk of disease incidence and spread in agrosystems [2,3,5].

The increasing demand for reuse of irrigation water by commercial nurseries and vegetable growers calls for generic on-site diagnostic strategies for early monitoring of a range of pathogens. High throughput sequencing approaches have made profiling of complex microbial communities in many ecological systems possible [10]. These techniques are culture-independent, highly sensitive and robust, and could enable the accurate detection and quantification of water-borne pathogens for quick diagnosis, as well as modelling and surveillance of microbial communities. The third-generation Oxford Nanopore sequencing instrument, MinION is portable, relatively cheap and offers rapid and accurate identification of pathogenic organisms [11–14]. In human diagnostics, the MinION has been used successfully for timely diagnosis during epidemics caused by viruses such as Ebola, Zika and SARS-COV2 [14–17]. Moreover, MinION has been successfully used to rapidly detect microbial pathogens in plants and insect pests tissues [18].

Although Oxford Nanopore technology (ONT) has great diagnostic potential, its usage in rapid detection of pathogens in greenhouse settings is still limited. Specifically, the use of the MinION device for metabarcode-based identification of fungi and oomycetes have been reported to produce very low proportions of meaningful sequences, frequent tag-switching events and highly unequal sequencing depth across samples [19]. However, current advances in ONT have been unprecedented, with continuous improvement in protocols for workflows, sequencing accuracy, basecalling, bioinformatics pipelines and reference databases [20,21]. Recently, ONT has been able to achieve >99% raw read accuracy using the most recent Kit V14 chemistry and R10.4.1 pore (https://nanoporetech.com/q20plus-chemistry). These developments have increased interest in the utilization of ONT for human disease diagnostics [22–24] as well as pathogens in agrosystems [18,25,26].

Nevertheless, the analysis of sequencing data derived from environmental samples remain challenging because of the complexity of microbial communities, combined with technical biases that are introduced during sample preparation, DNA extraction, polymerase chain reaction (PCR), library preparation, and downstream analysis [27–29]. Importantly, constraints

associated with taxonomic annotation due to incomplete reference databases and limited benchmarking of the quantitative performance of sequencing platforms hinder the reliability of these technologies in disease surveys [30,31]. To mitigate these challenges, reference standards including biological and artificially constructed spike-ins have been suggested and tested as internal reference standards [27,32]. Reference standards are added to the biological samples or DNA extracts prior to library preparation and further validation in the workflow procedure [28]. They enable accurate measurement of technical variables that could bias analysis, evaluate quantitative accuracy and reproducibility, and are useful for benchmarking bioinformatics pipelines [28,29,33,34]. Compared to biological mock communities, artificial spike-ins better assess analytical performance and data quality on a per-sample basis, and are used for absolute quantification or read count normalization, based on the known amount of spike-ins added to the samples [29,35].

The aim of this study was to assess the performance of a MinION sequencing approach in profiling microbial pathogens in greenhouse irrigation water systems based on fungal and oomycete amplicons. The objectives of our study were to i) examine the ability of MinION to assess the taxonomic composition of fungal and oomycetes communities in greenhouse irrigation water systems and ii) to assess the ability of phytopathogen quantification in greenhouse irrigation water systems using Oxford nanopore sequencing.

## Materials and methods

### Greenhouse water samples

For an initial assessment of the microbial composition of greenhouse water, we collected samples of recirculating irrigation water from cultures of tomato, cucumber and the ornamental *Aeschynanthus* grown in Denmark in 2020 in commercial greenhouses (Table S1 in S1 File). There were no notable disease symptoms in the plants recorded at the time of collection. Greenhouse water samples (1 L each) were collected from the water flow returning from plant cultures prior to entering the return tanks of the irrigation system at weekly intervals for the three first weeks of June for cucumber, from late June to early September for tomato and from late March to mid-November for *Aeschynanthus*. The collected samples were frozen at -20 ˚C until downstream processing.

### Water filtration and DNA extraction

The frozen water samples were thawed and filtered through 0.22 μm Sterivex™ sterile filters (Merck, Darmstadt, Germany). We extracted DNA from the filters using the DNeasy® Power-Water® Sterivex™ extraction kit (Qiagen, Hilden, Germany), following the manufacturer's instructions. The extracted DNA was quantified using a Qubit 3 Fluorometer and stored at -20 ˚C. A detailed description of the procedure and DNA concentrations (Table S1 in S1 File) can be found in the additional materials and methods (**see** S1 File).

### DNA amplification

We used ITS1-F/ITS4 [36,37] and ITS1oo/ITS4 primers [37,38] to amplify fungal and oomycete ITS regions, respectively. Fungal PCR was performed in a reaction mixture of 25 μl consisting of 1× PCR reaction buffer, 1.5 mM $MgCl_2$, 0.2 mM dNTPs, 1 μM of each primer, 1 U of GoTaq Flexi polymerase (Promega Corporation, Madison, USA), and 1 μl of DNA template. PCR was conducted in a GeneAmp PCR System 9700 thermal cycler (Thermo Fisher Scientific, Applied Biosystems, Foster City, CA, USA) with the following program: initial denaturation at 93 ˚C for 3 min, followed by 35 cycles at 95 ˚C for 1min, 60 ˚C for 1 min, 72 ˚C for 2

min, and a final elongation step at 72 ˚C for 10 min. Oomycete PCR was prepared in a 25 μl reaction consisting of 12.5 μl Platinum SuperFi Master Mix (Thermo Fisher Scientific, Waltham, MA, USA), 2 μM of each primer and 2 μl of DNA template, and amplified with the following program: initial denaturation at 95 ˚C for 15 min, followed by 35 cycles at 95 ˚C for 30 s, 55 ˚C for 30 s, 72 ˚C for 1 min, and a final elongation step at 72 ˚C for 10 min. After PCR, amplicons were confirmed by visualization in a 1.5% agarose gel using SYBR staining and quantified using the Qubit™ 1x dsDNA high sensitivity Assay kit (Qubit™ 3 Fluorometer, Thermo Fisher Scientific).

## MinION library preparation and sequencing

Amplicon library preparation was carried out according to the 1D PCR barcoding amplicons SQK-LSK109 protocol (Oxford Nanopore technology). Briefly, an end-repair was performed using the NEB-Next Ultra II End-prep reaction buffer and enzyme mix (New England Bio-Labs, Ipswich, MA), and samples were purified using MAGBIO Highprep beads (MagBio Genomics Inc, USA). The end-prep DNA from each sample were barcoded using the Native Barcoding Expansion kit (EXP-NBD196). Adapter ligation was done as recommended and the ligated DNA library was purified with MAGBIO Highprep, and finally eluted in 15 μL of Elution Buffer. Sequencing was performed on R9.4.1. flow cells on the MinIon MK1C and run for at least 22 hours.

## Data processing and analysis

Fast5 files were basecalled on MinION Mk1C using guppy with high accuracy model and read filtering of min_score = 8. Sequences were demultiplexed with guppy (https://timkahlke. github.io/LongRead_tutorials/BS_G.html) and adapters removed using porechop (https:// github.com/rrwick/Porechop). The trimmed sequences were then filtered based on quality and length using filtlong (https://github.com/rrwick/Filtlong). Finally, sequences were mapped against the UNITE reference database [39] using minimap2 [40] and summarized to operational taxonomic units (OTUs) in R. For a detailed description, see the S1 File. Fungal and oomycete community analysis was performed in R version 3.5.2 [41] primarily, using the ampvis2 [42], vegan (v2.5.7) [43], phyloseq (v1.34.0.) [44], and ggplot2 (v3.3.2) [45] packages. In addition, we performed a regression analysis to assess the relationship between spike-in reads and dilution series by using the stat_poly_eq function in the R package 'ggpmisc' (https://cran. r-project.org/web/packages/ggpmisc/index.html).

## Water-borne infection model

**Production of *Fusarium oxysporum* spores for inoculation.**   Mycelial plugs (3 plugs/ 100ml) from 7 days old cultures of *Fusarium oxysporum* f.sp. *mathioli* (FOM) were transferred into 1L sterile liquid carboxymethylcellulose (CMC) (high viscosity sodium salt, #C5013: Sigma Aldrich, St. Louis, MO, USA) broth containing 1.5% CMC-Na, 0.1% $NH_4NO_3$, 0.1% $KH_2PO_4$, 0.1% yeast extract and 0.05% $MgSO_4 \cdot 7H_2O$) and cultured for 3 days at 22˚C in the dark with orbital shaking (125 rpm) (Figure S1A, S1B in S1 File). Fungal spores were harvested by filtering through sterile Miracloth and the spores in the filtrate were concentrated by 15 minutes of centrifugation (4500 *g*) at room temperature. The spores were then washed twice with sterile deionized distilled $H_2O$, centrifuged at 7500 *g* for 5 min at room temperature and the supernatant discarded. The spore pellet was resuspended in sterile water and spore concentrations estimated using a hemacytometer placed under a Leica DM2000 microscope (Leica Microsystems GmbH, Wetzlar, Germany) (Figure S1C in S1 File).

**Experimental setup.** To assess whether ONT MinIon nanopore sequencing was able to detect water-borne fungal propagules before the onset of visible symptoms, we established a tomato-FOM pathosystem: we inoculated tomato plants with FOM (infection 'donor plants') and co-cultivated these symptomatic donor plants with non-infected plants ('receiver plants') in a hydroponic system. In this system, we monitored fungal and oomycete communities in the water by MinION amplicon sequencing, as well as the development of wilting and stunting symptoms of FOM infection in the receiver plants.

We inoculated the donor plants by dipping the roots of two-weeks old tomato seedlings grown in Grodan mineral wool (https://www.grodan.com) in a FOM spore suspension ($10^7$ spores ml$^{-1}$) (see above). Non-inoculated control plants were dipped in distilled water. After inoculation, the donor and control plants were transferred to new mineral wool cubes and incubated separately for two weeks in trays with fertilizer solution in a greenhouse. During this time, symptoms of FOM infection developed on the inoculated plants (Figure S1D in S1 File), whereas non-infected plants did not exhibit any disease symptoms. After the incubation period, we moved the plants into fresh trays (110x100x20 cm). Each tray contained two infected donor plants and one 2-weeks old receiver plant. Similarly, we included a control tray with three non-infected plants only. Each setup was replicated 7 times and maintained under greenhouse conditions. The volume of water in the trays (3 L) was kept constant throughout the duration of the experiment.

We monitored disease progression in the donor and receiver plants every second day. Water samples (250 ml) were collected at the start of the experiment, referred to at 0-day samples (T0) and subsequently at weekly intervals; 7-days (T1), 14-days (T2), 21-days (T3), 28-days (T4), 35-days (T5) and 42-days (T6). The samples were stored at -20 ˚C for later DNA extraction PCR and sequencing as described above.

## Spike-in amplification and quantitative assessment

**Biological Mocks.** Fungal isolates of *Fusarium oxysporum* f.sp. *mathioli* (FOM) [46], *Verticillium dahliae* (CBS 130341) and *Saccharomyces cereviasiae* (common baker's yeast) and isolates of the oomycetes *Phytophthora infestans* (EU41), *P. fragariae* (CBS 209.46) and *Globisporangium intermedium* (previously *Pythium intermedium*) (CBS 102.607) were used in this study. Isolates of fungi and oomycetes were maintained on potato-dextrose agar (PDA) and V8 media, respectively. DNA from the cultures was extracted using the DNeasy Plant Mini kit (Qiagen, Hilden, Germany) according to the manufacturer's instructions. Isolated DNA was quantified and stored at – 20 ˚C until used for PCR.

The fungal mock (Fmock) consisted of equal amounts of DNA from *Fusarium oxysporum*, *Verticillium dahliae* and *Saccharomyces cereviasiae* (Table S2 in S1 File). The oomycetes mock (Omock) was constructed using equal amounts of DNA from *Phytophthora infestans*, *P. fragariae* and *G. intermedium*. We obtained a final concentration of 1.75 ng/ul and 0.84 ng/ul for the Fmock and Omock, respectively.

**Artificial spike-in design and preparation.** Artificial spike-ins were used to assess the performance and quantification of fungal and oomycete pathogens using the MinION. We created artificial sequences with characteristics (length and nucleobase distribution) similar to the ITS regions that we amplified with the primer pairs targeting fungi and oomycetes, respectively. To create artificial sequences, we downloaded fungal *F. oxysporum* (accession number: MT453296.1) and *Saccharomyes cerevisiae* (accession number: MK649847.1) sequences that had been amplified by the ITS1-F/ITS4 primers and the oomycete *Phytophthora infestans* (accession number: EF126351.1) and *Pythium* sp. (accession number: MW366735.1) sequences that had been amplified by ITS1oo/ITS4 primers, from the NCBI database (S1 File).

The ITS1-F/ITS4 and ITS1oo/ITS4 amplifiable regions of these sequences were then mirrored to create non-sense DNA sequences as described by Blackburn et al. [28], with no likely identity to fungal or oomycete sequences, respectively, while primer binding sites were replaced with complementary sequences of the respective primers. A BLAST search of the full length ITS artificial sequences confirmed no significant similarity to any sequence in the NCBI database. The artificial sequences or spike-ins were synthesized and cloned into pUC19 using standard methods (Thermo Fisher Scientific, Carlsbad, USA). In total, we created 4 artificial spike-ins derived sequences from FOM (AsFo), *Saccharomyes cerevisiae* (AsSc), *Phytophthora* (AsPy) and Pythium (AsPy).

The pUC19 plasmid cloning vector with spike-in sequence inserts were transformed into electro-competent *Escherichia coli* stellar cells following the manufacturer's instructions (Takara Bio Company, CA, USA). We extracted plasmid DNA from overnight cultures using the MACHEREY-NAGEL miniprep kit (Düren, Germany) and confirmed the artificial sequences by Sanger sequencing. Positive spike-in plasmids were linearized using the *Pvu*I restriction enzyme according to the manufacturer's instructions (New England BioLabs, Ipswich, MA). The integrity and sizes of the linearized plasmids were verified by electrophoresis using 1% agarose gel and EtBr staining. The resulting spike-in fragments were purified using NucleoSpin kit (MACHEREY-NAGEL, Düren, Germany). Plasmid DNA concentrations were determined using Nanodrop 1000 spectrophotometer (Thermo Scientific) and then stored at −80C.

**Experiment 1:** To establish optimal spike-in concentrations for quantitative assessment during MinION sequencing, we performed serial dilutions of the spike-ins into DNA extracted from the tomato greenhouse water with a concentration of 0.09 ng/ul as a background. 'Fungal' and 'oomycete' artificial spike-in concentrations were adjusted to 1 ng/ul before serial dilutions ($6 \times 10$-fold) in the greenhouse water DNA. We amplified the spiked samples following the protocols outlined above and prepared amplicon libraries, which were sequenced using the MinION MK1C as described above.

**Experiment 2:** Based on the artificial spike-in sequence read outputs from the serially diluted samples (Figure S2 in S1 File), we set up a second experiment with the objective to assess whether sequence reads of target organisms (Fmock and Omock) would also reflect quantitative variations in the target abundance. The Fmock and Omock were serially diluted ($6 \times 10$-fold) into the background greenhouse water DNA sample with a DNA concentration of 1.56 ng/ul (Figure S3A, S3B in S1 File). To be within a well performing spike-in range (Table S3 in S1 File), we constructed two Fmocks with artificial spike-ins at two different dilutions, i.e., AsFmock1with AsFo:$10^{-5}$ and AsSc:$10^{-3}$, and AsFmock2 with AsFo:$10^{-6}$ and AsSc:$10^{-4}$. Likewise, we constructed two Omocks, i.e., AsOmock1 with AsPh:$10^{-2}$ and AsPy:$10^{-3}$, and AsOmock2 with AsPh:$10^{-3}$ and AsPy:$10^{-4}$ (Figure S3A, S3B in S1 File). We added 1 uL of the selected spike-in sample to each dilution of the mock samples. The setup also included a positive control series, i.e., a $7 \times 10$-fold serially diluted biological mock only with no spike-ins added and a negative control, i.e., the background greenhouse water DNA samples. All treatments were replicated three times.

**Experiment 3:** The outcomes of Experiment 1 and 2 suggested that the DNA concentration in the sample background significantly affected the performance of artificial spike-ins. To test this hypothesis, we serially diluted oomycete artificial spike-ins into greenhouse water DNA samples (drawn from the same original sample) with three different concentrations: low (0.09 ng/ul), medium (0.5 ng/ul) and high (1 ng/ul). All treatments were replicated three times. The DNA concentrations were quantified prior to amplification, library preparation and MinION sequencing as previously described.

## Result and discussion

The high prevalence of fungal and oomycete diseases in greenhouse crop production, orchestrated by the intensive cropping system and also by the reuse of water, requires novel diagnostics strategies for timely management and control. In the present study, we tested the portable and relatively simple and cheap ONT MinION sequencing device for early identification and quantification of fungal and oomycete pathogens in irrigation water samples. We assessed the performance of the MinION using both biological and synthetic standards consisting of fungal and oomycete DNA via native barcoding and full-length ITS amplicon sequencing.

### Sequence data characteristics

Native barcoding of amplified DNA of fungal and oomycete origin and full-length ITS sequencing yielded varying number of sequence reads (range: 20471–1738404) from each MinION run. A summary of sequences obtained after basecalling and quality filtering from each MinION sequencing including all samples are summarized (Tables S4-S8 in S1 File). Sequence data are available in the Sequence Read Archive (SRA) database at NCBI under Bio-Project ID PRJNA1063628 with the accession numbers *SUB14149061*, *SUB14149186*, *SUB14152390*, *SUB14152424*, *SUB14144130* and *SUB14152446*.

**Profiling microbial communities in greenhouse water using Oxford nanopore.**   First, we profiled irrigation water DNA samples from different greenhouse cultures, amplified with fungal or oomycete primers, by MinION sequencing. The relative abundances of reads originating from fungi and oomycetes were distinct in the tomato, cucumber and *Aeschynantus* irrigation water samples (Fig 1). The fungal genera *Candida*, *Russula*, *Tomentella* and *Aspergillus*, and the oomycete genera *Pythium*, *Lagenidium*, *Sphaerothecum* and *Paraphysomonas* were

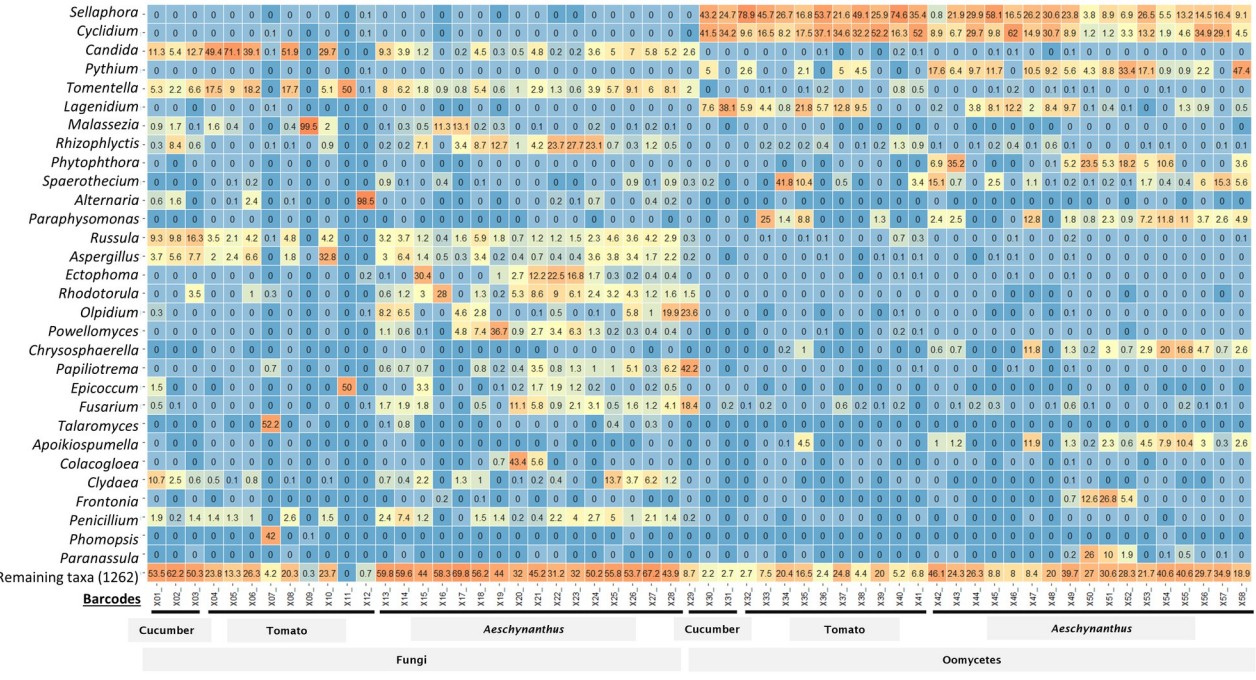

**Fig 1. Assessment of fungal and oomycetes communities in greenhouse irrigation water using Oxford nanopore sequencing.** Relative abundances of fungal and oomycetes communities in greenhouse irrigation water sampled from tomato, cucumber and *Aeschynanthus* production sites in Denmark in 2020. Taxonomic classification at genus level is shown in the y-axes, while samples represented by barcode number are indicated in the x-axis.

dominant across the samples, although there were differences. The relative abundances of *Rho-dotorulla*, *Olpidium*, *Ectophoma* and *Phytophthora*, for instance, were higher in the irrigation water from the *Aeschynantus* cultures compared with the water from tomato and cucumber. We found that the oomycete primers that we used also amplified DNA from other protist groups, notably *Sellaphora* and *Cyclidium*, probably due to their close relatedness to oomycetes [47]. This amplification of non-target organisms can probably be resolved using more specific primers targeting only oomycetes.

The greenhouses from where samples were taken had no records of fungal and oomycete disease at the time of collection. Therefore, to test the ability of the MinION in detecting pathogens in irrigation water samples from greenhouses with ongoing disease outbreaks, we established a tomato-*Fusarium* pathosystem including F*usarium*-infected donor plants and healthy recipient plants to explore *Fusarium* detection and quantification at different time points via sequencing of DNA from the irrigation water. We were able to detect *Fusarium* in water samples from trays with donor plants shortly after transferring infected donor plants into the trays (Figure S4 in S1 File). This result suggest that the MinION sequencing could be ideal for early detection and quantification of fungal pathogens in water (Fig 2). In support of this, symptoms of wilting caused by *FOM* were observed in the tomato receiver plants after 3 weeks of co-cultivation with donor plants (Figure S1D in S1 File). We expected a continuous release of spores from the donor plants which would increase the amount of *Fusarium* spores in the irrigation water. Eventually, these spores would infect the healthy plants with increasing time of exposure. We found a high abundance of *Fusarium* in the earliest time point of 7 days after cocultivation but following, the abundance significantly declines (Tray 1; $R^2$adj = 0.23, padj < 0.025 and Tray 2: $R^2$adj = 0.24, padj < 0.014) with time (Fig 2). The early detection of *Fusarium* is significant with respect to the applicability of MinION for the timely identification of fungal pathogens in irrigation water. The observed decline in *Fusarium* abundance at later stages could possibly be caused by the poor survival of *Fusarium* spores in stagnant water [48], which was the case for the water in our set up.

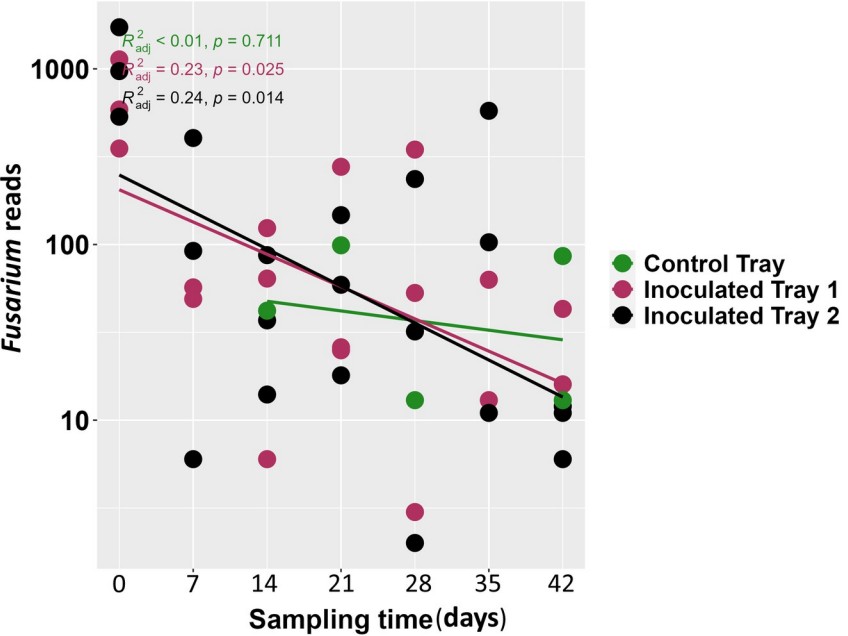

**Fig 2. *Fusarium* reads obtained from water mediated experiment at different sampling time.**

**Assessing the biological mocks and artificial spike-ins as standards using our Oxford Nanopore approach.** Next, we tested the ability of the MinION sequencing platform to detect artificial spike-ins and biological mock communities. Routinely, biological and artificial standards have been spiked in various biological samples to test the analytical performance of sequencing platforms such as the Oxford nanopore and Illumina [27–29]. An important consideration when adding standards is the need to establish a suitable working concentration to avoid masking the signal from the actual sample content. Besides, it is important to ensure that the spike-ins are easily distinguishable from the microbial diversity of the biological samples [49]. Our artificial spike-ins were designed by mirroring existing biological sequences [28] of fungal and oomycete ITS sequences, respectively, thus, sharing the same nucleotide composition and repetitiveness of the individual species, and at the same not resembling sequences of organisms in the sample that would be amplified with the primers used.

In Experiment 1, we serially diluted spike-ins into a greenhouse water background DNA. This test confirmed that all spike-ins were detected via MinION amplicon sequencing, and that the relative abundance of spike-in DNA declined with increasing dilution (Fig 3). Because of the increasing dilution of the spike-ins, we observed higher relative abundances of reads coming from the water background (Fig 3). In the samples spiked with artificial fungal-derived spike-ins (AsFo and AsSc), the AsFo was detected in all samples with declining read counts in the diluted samples. However, the relationship between the actual spike-in dilutions and the relative counts of the spike-in read output was not linear (Figure S2 in S1 File). The AsSc artificial spike-in was more sensitive to dilution and could only be detected down to $10^{-3}$ x dilution (exp-3). Similarly, the oomycete spike-ins AsPh and AsPy were only detected in the undiluted and $10^{-1}$ samples. Both fungal and oomycete spike-ins were added to the same background DNA sample, and we speculate that the observed differences in detection thresholds across dilution series of the artificial spike-ins could be attributed to amplification biases in biological mixtures [50]. The length of fungal amplicons produced using ITS1F and ITS4 primers vary considerably, typically from 420 to 825 bp [51]. In this study, the fungal spike-in AsFo with the shortest length (538 bp) yielded the highest number of sequence reads in all the dilution series (Fig 3A and 3B). In conclusion, we recommend the addition of several spike-ins or the use of

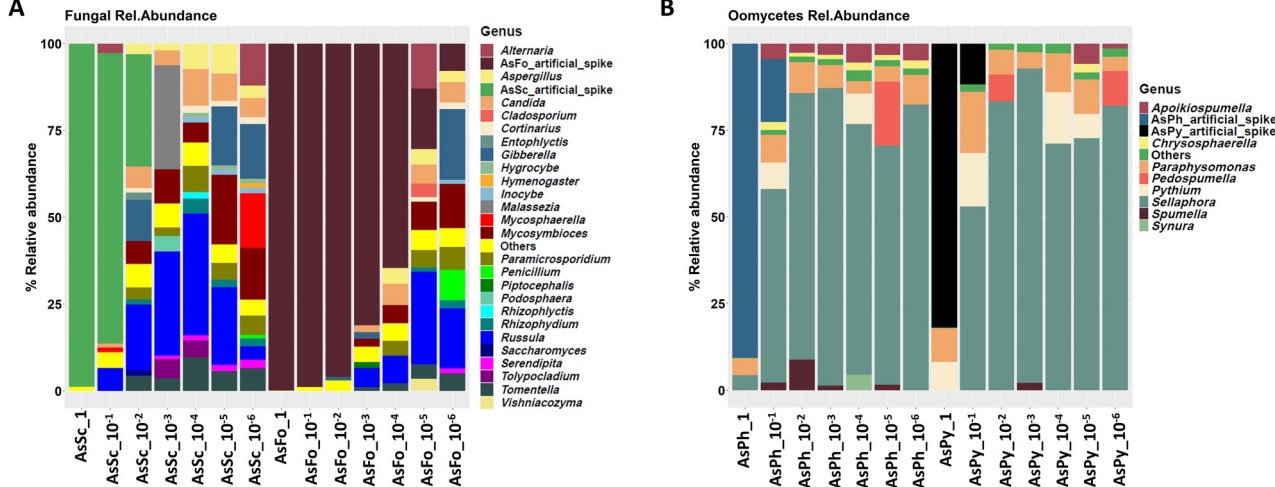

**Fig 3. Assessment of serially diluted artificial fungal (AsFo and AsSc) and oomycetes (AsPh and AsPy) spike-ins spiked into greenhouse DNA of concentration of 0.09 ng/ul.** A) Relative abundance of fungi (at genus level) and artificial spike-ins in 6× 10-fold dilution samples. B) Relative abundance of oomycetes (at genus level) and artificial spike-ins in 6× 10-fold dilution samples.

shorter sequence fragments in constructing artificial spike-ins for quantitative metabarcoding assessment when using MinION sequencing. Moreover, copy number variation among fungal isolates could account for the read abundance differences between fungi, as this would affect both PCR amplification and sequencing [52,53].

After confirming that the artificial spike-in DNAs were detectable via MinION amplicon sequencing, we assessed their performance in background samples from greenhouse water having varying DNA quantities in Experiment 2. Biological samples are generally characterized by varying levels of microbial loads, and the dynamic detection range of artificial reference standards should be able to cover this variation. Thus, the varying concentrations of the background samples would enable us to assess the quantitative performance of the spike-ins in the different water samples.

The addition of fungal (FOM, *Verticillium dahlia* and *Saccharomyces cereviasiae*) and oomycete (*Phytophthora infestans*, *P. fragariae* and *Globisporangium intermedium*) mock communities in different concentrations (dilution series) to the greenhouse background DNA demonstrated that read outputs of the mock taxa reflected the decreasing concentration of mock community DNA in the serial dilutions (Fig 4A and 4B). This finding demonstrates the ability of the MinION to relatively quantify microorganisms, including plant pathogens, in biological samples having different microbial loads. As expected, we found that the relative abundance of unclassified taxa as well as major fungal genera including *Rhyzophlyctis*, *Candida*, *Tomentella*, and *Cladosporium* increased with increasing Fmock dilutions. These taxa were higher in the background water DNA, thus highly representative in the highly diluted samples, and were only outcompeted in samples with higher quantities of the biological mocks. *Phytophthora* and *Pythium* dominated the oomycete communities, while the remaining taxa were extremely low in abundance (Fig 4B). The relative abundances of *Phytophthora* (*P. infestans* and *P. fragariae*) were generally higher than *Pythium*, and could be caused by

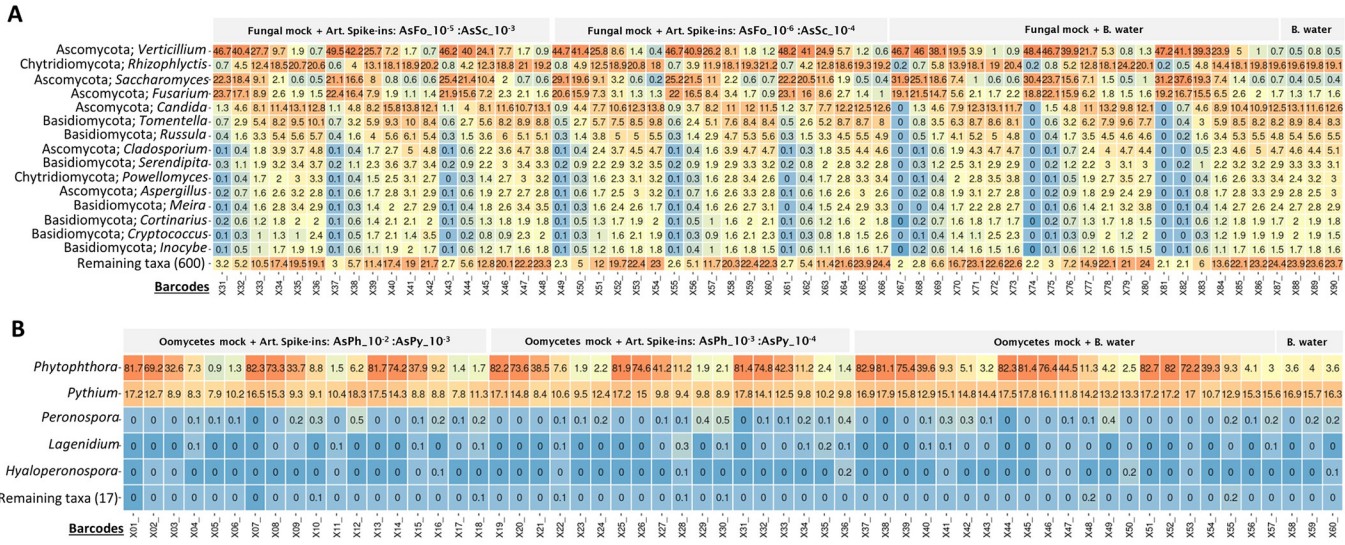

**Fig 4. Heatmap of fungal and oomycetes relative abundances across 6x 10-fold serially diluted biological mock samples.** Fungal mock communities consisted of DNA from *Fusarium oxysporum* f.sp. *mathioli*, *Verticillium dahliae* and *Saccharomyces cereviasiae*, and oomycete mock communities consisted of DNA from *Phytophthora infestans*, *P. fragariae* and *G. intermedium*. A) Heatmap of fungal relative abundances of greenhouse DNA samples spiked with serially diluted (6x 10-fold) fungal mock and 1 ng/ul of fungal artificial spike-ins (AsFo and AsSc). B) Heatmap of oomycete relative abundances of greenhouse DNA samples spiked with serially diluted (6x 10-fold) oomycetes mock and 1 ng/ul of oomycetes artificial spike-ins (AsPh and AsPy). Taxonomic classification at phylum and genus levels are shown in the y-axes, while samples represented by barcode number are in indicated in the x-axis.

either DNA extraction efficiency, amplification efficiency or both factors. Altogether, results of the distinct microbial taxa abundances in samples with varying concentrations further depict the suitability of the MinION in monitoring microbial communities in greenhouse production.

In Experiment 1, we retrieved much higher numbers of artificial spike-in reads in samples with lower background DNA concentration, whereas in experiment 2 (where we tested the spike-ins in biological samples) the artificial spike-ins were not detectable in any of the diluted samples. This finding led us to hypothesize that the sample DNA concentration could directly affect spike-in performance. Previous studies have shown that the addition of excessive spike-in material resulted in the disappearance of biological sample reads [54]. Artificial spike-ins constructed using the mirrored approach retains the same properties of the original sequence including nucleotide composition and repetitiveness and thus performs well in steps such as PCR amplification [28]. However, the poor performance of such spike-ins may arise from the quantity added and the type of biological mock used [28].

To test whether the concentration of the background DNA from greenhouse water affected the performance of the artificial spike-ins (experiment 3), we added serially diluted oomycete spike-ins into greenhouse water DNA samples with three different concentrations: low (0.09 ng DNA/ul), medium (0.5 ng/ul) and high (1 ng/ul). Our results show that the diluted artificial spike-ins were detectable in all samples, and the number of spike-in derived reads was highest in the background samples with the lowest DNA concentration (Fig 5A). In addition, there were notable differences in the relative abundances of the most abundant oomycete genera across the serially diluted samples within the tested background DNA concentrations. Similarly, the abundances of *Phytophthora* and *Pythium* artificial spike-ins generally decline with increasing dilutions (Fig 5B). Our results thus suggest that artificial spike-in performance has a narrow working range that may preclude the use of such references in the system.

In conclusion, our results suggest MinION as a promising sequencing tool for monitoring of pathogens in greenhouse irrigation water. We utilized the MinION for profiling both oomycete and fungal communities in water samples collected from greenhouses, *FOM*-inoculated water, and water samples spiked with oomycetes and fungal pathogens. MinION sequencing

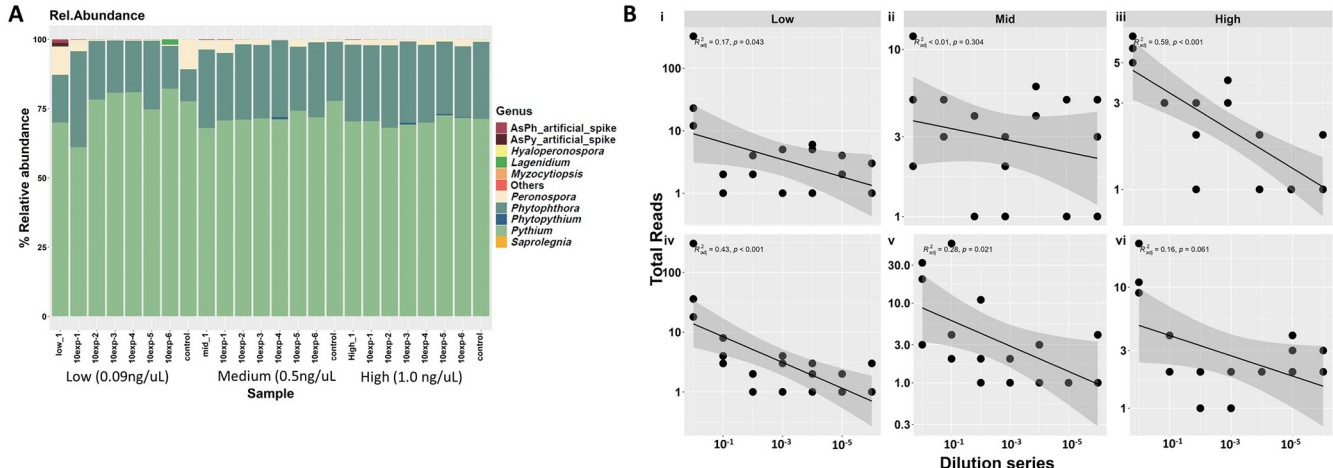

**Fig 5. Assessment of artificial oomycetes spike-ins (AsPh and AsPy) diluted serially into different background DNA concentrations: Low (0.09ng/uL), Medium (0.5ng/uL) and High (1.0 ng/uL).** A) Relative abundance profile of oomycetes (at genus level) and artificial spike-ins in samples with different DNA concentrations. B) Total reads of oomycetes artificial spike-ins in serially diluted samples with low, medium and high background DNA concentrations: *Phytophthora* artificial spike-ins (i, ii, and iii) and *Pythium* artificial spike-ins (iv, v and vi).

produced distinct microbial profiles in the irrigated water samples and in the *FOM* inoculated water samples. A limitation to our study is that we could not identify fungi and oomycetes to species level, partly due to the incompleteness of reference databases, as previous studies have also reported [55]. Fungal and oomycete mock analysis yielded distinct abundances of added fungal and oomycete DNA corresponding to their respective serial dilutions. We found that artificial spike-ins were highly affected by background DNA concentrations, and this affected sequence read numbers obtained from the MinION sequencing. The spike-in reads were uneven with respect to dilution series and were least retrievable in the background samples having the highest DNA concentration, suggesting a narrow range of performance. Also, the shortest artificial spike-in (*F. oxysporum* artificial spike-in: 538bp) was the most recoverable in a broad range of serial dilutions. We observed that the spike-ins with shorter length (538bp) generally gave reads in most of the dilutions compared with the longer spikes (>790bp). Based on this finding, we recommend short-length artificial spike-ins (< 538bp), but more studies are needed to test a broader range of artificial spike-ins for applications in quantitative MinION sequencing. While we acknowledge the prospect of ONT MinION sequencing in disease diagnostics, a continual benchmarking of the MinION, specifically, with improved nanopore flowcells, chemistries and basecallers will pave way for quantitative metabarcoding pipelines, thus improving future plant disease diagnostics.

## Supporting information

**S1 File. This file contains additional details of materials and methods, supplementary figures 1–4 and tables 1–8.**
(PDF)

## Acknowledgments

We thank Thanassis Zervas for initial sequencing support and Zelalem E. Bekalu. for cloning assistance. We also thank Mathilde Schiøtt Dige and Simone Ena Rasmussen for their excellent laboratory assistance.

## Author Contributions

**Conceptualization:** Enoch Narh Kudjordjie, Mai-Britt Brøndum, Mads Grønvald Johnsen, Mogens Nicolaisen, Mette Vestergård.

**Data curation:** Enoch Narh Kudjordjie, Anne Saaby Schmidt-Høier, Mads Grønvald Johnsen.

**Formal analysis:** Enoch Narh Kudjordjie, Anne Saaby Schmidt-Høier.

**Funding acquisition:** Mai-Britt Brøndum, Mads Grønvald Johnsen, Mogens Nicolaisen, Mette Vestergård.

**Investigation:** Enoch Narh Kudjordjie, Anne Saaby Schmidt-Høier, Mai-Britt Brøndum, Mads Grønvald Johnsen, Mogens Nicolaisen, Mette Vestergård.

**Methodology:** Enoch Narh Kudjordjie, Anne Saaby Schmidt-Høier, Mai-Britt Brøndum, Mads Grønvald Johnsen, Mogens Nicolaisen.

**Project administration:** Mai-Britt Brøndum, Mads Grønvald Johnsen, Mogens Nicolaisen, Mette Vestergård.

**Resources:** Anne Saaby Schmidt-Høier, Mai-Britt Brøndum, Mads Grønvald Johnsen, Mogens Nicolaisen.

**Software:** Enoch Narh Kudjordjie, Anne Saaby Schmidt-Høier, Mai-Britt Brøndum, Mads Grønvald Johnsen.

**Supervision:** Mai-Britt Brøndum, Mads Grønvald Johnsen, Mogens Nicolaisen, Mette Vestergård.

**Validation:** Enoch Narh Kudjordjie, Anne Saaby Schmidt-Høier, Mogens Nicolaisen, Mette Vestergård.

**Visualization:** Enoch Narh Kudjordjie, Anne Saaby Schmidt-Høier, Mette Vestergård.

**Writing – original draft:** Enoch Narh Kudjordjie.

**Writing – review & editing:** Enoch Narh Kudjordjie, Anne Saaby Schmidt-Høier, Mogens Nicolaisen, Mette Vestergård.

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
