## [Decision Letter · Decision Letter 0]

2 Jan 2024

PONE-D-23-39732Early assessment of fungal and oomycete pathogens in greenhouse irrigation water using Oxford nanopore amplicon sequencingPLOS ONE

Dear Dr. Kudjordjie,

Thank you for submitting your manuscript to PLOS ONE. After careful consideration, we feel that it has merit but does not fully meet PLOS ONE’s publication criteria as it currently stands. Therefore, we invite you to submit a revised version of the manuscript that addresses the points raised during the review process.

We look forward to receiving your revised manuscript.

Kind regards,

Jaime E. Blair, PhD

Academic Editor

PLOS ONE

Journal Requirements:

"GUDP (project no. 32002)" 

4. Please expand the acronym “GUDP” (as indicated in your financial disclosure) so that it states the name of your funders in full.

5. Thank you for stating the following in the Acknowledgments Section of your manuscript: "The research was funded by GUDP (project no. 32002)."

Please remove any funding-related text from the manuscript and let us know how you would like to update your Funding Statement. Currently, your Funding Statement reads as follows: "GUDP (project no. 32002)" 

6. In the online submission form, you indicated that Data presented in this study are available upon request.

Additional Editor Comments :

Please specifically note the comments by Reviewer #2 in your revision, and update taxonomic names as appropriate.

Reviewers' comments:

Reviewer's Responses to Questions

**Comments to the Author**

1. Is the manuscript technically sound, and do the data support the conclusions?

Reviewer #1: Yes

Reviewer #2: Yes

2. Has the statistical analysis been performed appropriately and rigorously? 

Reviewer #1: Yes

Reviewer #2: Yes

3. Have the authors made all data underlying the findings in their manuscript fully available?

Reviewer #1: No

Reviewer #2: Yes

4. Is the manuscript presented in an intelligible fashion and written in standard English?

Reviewer #1: Yes

Reviewer #2: Yes

5. Review Comments to the Author

Reviewer #1: The manuscript reads very well and I had an easy time getting through it. I just have a few minor comments for authors (see below).

Page 6 and 7 under Data processing and analysis, I suggest authors cite filling, porechop, guppy, minimap2, or at least provide a link to where software could be downloaded.

Page 9. Add the species, so consider changing "...the oomycete Phytophthora (accession number:

EF126351.1) and Pythium (accession number: MW366735.1)..." To "...the oomycete Phytophthora infestans (accession number: EF126351.1) and Pythium sp. (accession number: MW366735.1)"

Pythium intermedium was transferred to Globisporangium intermedium. I suggest authors use the new name (see Uzuhashi et al. 2010 "Phylogeny of the genus Pythium and description of new genera" and Nguyen et al. 2022 "Whole genome sequencing and phylogenomic analysis show support for the splitting of genus Pythium"

I suggest italicizing the genus names in the figures, if not too much of a hassle.

I suggest authors deposit raw data generated into a repository like NCBI SRA, and state the accession number in the manuscript.

Reviewer #2: This manuscript presents early assessment potential of used of MinION amplicon sequencing for environmental monitoring in irrigation water. They use spike material in irrigation water and evaluate the tool for identification of Fungi and Oomycete pathogens. I think this is valuable work and information that is need to evaluate the tools, however I think they can go more far in the analysis of it and state more about the finding of non spike material. They evaluate some potential but not too clear what the best dilution or way to do it more far, they did not made the analysis to the species level but only to the genera level.

Specific comments:

Abstract: I think they can add more information in abstract about type of samples, numbers more details on results. Region of amplicon used...

P3L69, should rewrite this sentence, a verb I think is missing.

P5 Greenhouse samples and material should be presented in a table to get more clear.

L150 need # for the supplement info in.

L157 any info about the DNA conc. used.

L204, info on Miracloth

L246, please check new litterature about Pythium intermedium, now Globisporangium and could be a complex species.

L258, the spike-in preparation not always to clear to follow, the plasmid and not plasmid prep ...

L299 Concentration of DNA, not always clear?

L304-305 on water or plasmid material, the 1.56ng/ul?

L325, seems results and discussion are put together in what I seen in https://journals.plos.org/plosone/s/submission-guidelines seems they are supposed to be distinct sections?

L376 and Figure 2 are very low correlation.

L410- seem you only have genera information, why and it is possible to get to the species level, would have been great information on the use of the method and limit. I believe information need to be add and also part of your limitation in your bioinformatic pipeline, should be part of discussion. Figure 3B do we see Phytophthora?

L449 I believe you want to say Phytophthora fragaria and not Pythium intermedium here in parenthesis. L461 more info about the amplification bias and/or references.

L461, interesting but I believe more work need to be done in analysis, info about the species level, any other info from the water samples, important species find, concern any info about the other genera found, any concern, any problem they could cause...Return from L347... Peronospora...

Figure 1 the names of plant hosts and Fungi and Oomycete do not necessary align with the samples position? Recheck it.

Recommendation on concentration what type of spiking, info sequences of water without spike material only?

6. PLOS authors have the option to publish the peer review history of their article (what does this mean?). If published, this will include your full peer review and any attached files.

Reviewer #1: No

Reviewer #2: No

---

## [Author Response · Author response to Decision Letter 0]

2 Feb 2024

Reviewer #1: The manuscript reads very well and I had an easy time getting through it. I just have a few minor comments for authors (see below).

Page 6 and 7 under Data processing and analysis, I suggest authors cite filling, porechop, guppy, minimap2, or at least provide a link to where software could be downloaded.

Response

We have provided links to all packages as suggested. 

Page 9. Add the species, so consider changing "...the oomycete Phytophthora (accession number:

EF126351.1) and Pythium (accession number: MW366735.1)..." To "...the oomycete Phytophthora infestans (accession number: EF126351.1) and Pythium sp. (accession number: MW366735.1)"

Response

We have added the species name as suggested.

Pythium intermedium was transferred to Globisporangium intermedium. I suggest authors use the new name (see Uzuhashi et al. 2010 "Phylogeny of the genus Pythium and description of new genera" and Nguyen et al. 2022 "Whole genome sequencing and phylogenomic analysis show support for the splitting of genus Pythium" We thank the reviewer for this update. We have changed the name from Pythium intermedium to Globisporangium intermedium.

I suggest italicizing the genus names in the figures, if not too much of a hassle.

Response

We have italized the genus names as suggested.

I suggest authors deposit raw data generated into a repository like NCBI SRA, and state the accession number in the manuscript.

Response

The raw sequence files have been deposited in NCBI SRA under the bioproject # PRJNA1063628.

Reviewer #2: This manuscript presents early assessment potential of used of MinION amplicon sequencing for environmental monitoring in irrigation water. They use spike material in irrigation water and evaluate the tool for identification of Fungi and Oomycete pathogens. I think this is valuable work and information that is need to evaluate the tools, however I think they can go more far in the analysis of it and state more about the finding of non spike material. They evaluate some potential but not too clear what the best dilution or way to do it more far, they did not made the analysis to the species level but only to the genera level.

Specific comments:

Abstract: I think they can add more information in abstract about type of samples, numbers more details on results. Region of amplicon used...

Response

We have edited the abstract as suggested. 

P3L69, should rewrite this sentence, a verb I think is missing.

Response 

We did not detect any mistakes

P5 Greenhouse samples and material should be presented in a table to get more clear.

Response

We have provided a table to give an overview as suggested in the supplementary table S1.

L150 need # for the supplement info in.

Response

We have revised the text. 

L157 any info about the DNA conc. used.

Response

We have provided the DNA concentrations in the supplementary Table S4.

L204, info on Miracloth. 

Response

It is a common autoclavable material used in microbiology for filtration of e.g. spore suspensions. 

L246, please check new litterature about Pythium intermedium, now Globisporangium and could be a complex species.

Response

We thank the reviewer for this update. We have changed the name from Pythium intermedium to Globisporangium intermedium.

L258, the spike-in preparation not always to clear to follow, the plasmid and not plasmid prep ...

Response

We have added a detailed reference for the plasmid preparation. The described method used follows the protocol by Blackburn et al., 2019. Doi: http://dx.doi.org/10.1038/s41596-019-0175-1.

L299 Concentration of DNA, not always clear?

Response

We have provided the DNA concentrations of all samples used in each experiment for MinIon sequencing in the supplementary Table S2 and S4.

L304-305 on water or plasmid material, the 1.56ng/ul?

Response

The DNA concentration of the water was 1.56ng/ul. The plasmid mocks were serially diluted, thus they have respective serial dilution concentrations (Supplementary Table S4). 

L325, seems results and discussion are put together in what I seen in https://journals.plos.org/plosone/s/submission-guidelines seems they are supposed to be distinct sections?

Response

We had several experiments, where new experiments were included to address questions raised by the preceding experiments. We believe that the merged results and discussion section best reflects this flow of reasoning and makes reading much easier. 

L376 and Figure 2 are very low correlation. 

Response

Fusarium reads significantly decline with increasing time in both Tray1 and tray 2, while the control tray was expectedly non-significant.

L410- seem you only have genera information, why and it is possible to get to the species level, would have been great information on the use of the method and limit. I believe information need to be add and also part of your limitation in your bioinformatic pipeline, should be part of discussion. 

Response

The main challenge of fungal species level resolution is the incompleteness of reference databases limiting the accurate and precise identification (species level) of fungal taxa, as previously described Ohta et al. 2023 doi: https://doi.org/10.1038/s41598-023-37016-0). We have added this limitation to the text in the manuscript. 

Figure 3B do we see Phytophthora? 

Response

Phytophthora was detected in very low abundance (< 0.01). Only taxa with relative abundance of > 0.01 are shown in Figure 3B. 

L449 I believe you want to say Phytophthora fragaria and not Pythium intermedium here in parenthesis. 

Response

We thank the reviewer for his oversight. We have corrected the name to P. fragariae in the text.

L461 more info about the amplification bias and/or references.

Response

The observation described in lines 469-479 led us to speculate an amplification bias as previously reported by Chen et al., 2016 (referenced accordingly).

L461, interesting but I believe more work need to be done in analysis, info about the species level, any other info from the water samples, important species find, concern any info about the other genera found, any concern, any problem they could cause...Return from L347... Peronospora...

Response

Zooming-in into fungal species is challenging, primarily due to the quality of current databases as mentioned in my earlier response. However, we were able to profile the targeted species and spike-ins used in the study using the MinIon sequencing. 

Detailed information about the water samples used are provided in supplementary Table S1. 

Regarding concerns of other species, we address particularly the high abundance of Sellaphora and Cyclidium in Lines 367-368.

Figure 1 the names of plant hosts and Fungi and Oomycete do not necessary align with the samples position? Recheck it.

Response

We thank the reviewer for pointing out. We have edited figure 1 by aligning sample names, positions to fungi and oomycetes categories. 

Recommendation on concentration what type of spiking, info sequences of water without spike material only?

Response

Although we used relatively few spikes and it is thus difficult to draw firm conclusions, we observed that spikes with shorter lengths gave reads in most dilutions compared with the longer spikes. Therefore, we recommend short-length spikes. We also observed that the background concentration affects spike-in performance. Future studies are therefore needed to test several artificial spikes for benchmarking quantitative MinION sequencing.

---

## [Decision Letter · Decision Letter 1]

27 Feb 2024

Early assessment of fungal and oomycete pathogens in greenhouse irrigation water using Oxford nanopore amplicon sequencing

PONE-D-23-39732R1

Dear Dr. Kudjordjie,

We’re pleased to inform you that your manuscript has been judged scientifically suitable for publication and will be formally accepted for publication once it meets all outstanding technical requirements.

Kind regards,

Jaime E. Blair, PhD

Academic Editor

PLOS ONE

Additional Editor Comments (optional):

Reviewers' comments:

Reviewer's Responses to Questions

**Comments to the Author**

1. If the authors have adequately addressed your comments raised in a previous round of review and you feel that this manuscript is now acceptable for publication, you may indicate that here to bypass the “Comments to the Author” section, enter your conflict of interest statement in the “Confidential to Editor” section, and submit your "Accept" recommendation.

Reviewer #1: All comments have been addressed

Reviewer #2: All comments have been addressed

2. Is the manuscript technically sound, and do the data support the conclusions?

Reviewer #1: Yes

Reviewer #2: Yes

3. Has the statistical analysis been performed appropriately and rigorously? 

Reviewer #1: N/A

Reviewer #2: Yes

4. Have the authors made all data underlying the findings in their manuscript fully available?

Reviewer #1: Yes

Reviewer #2: Yes

5. Is the manuscript presented in an intelligible fashion and written in standard English?

Reviewer #1: Yes

Reviewer #2: Yes

6. Review Comments to the Author

Reviewer #1: Thank you for addressing my comments.

Reviewer #2: They have respond to my comments. L189-205 information on software version should be add mainly for porechop and reference material.

7. PLOS authors have the option to publish the peer review history of their article (what does this mean?). If published, this will include your full peer review and any attached files.

Reviewer #1: No

Reviewer #2: No

---

## [Editor Report · Acceptance letter]

7 Mar 2024

PONE-D-23-39732R1 

PLOS ONE

Dear Dr. Kudjordjie, 

I'm pleased to inform you that your manuscript has been deemed suitable for publication in PLOS ONE. Congratulations! Your manuscript is now being handed over to our production team.

Kind regards, 

on behalf of

Dr. Jaime E. Blair 

Academic Editor

PLOS ONE